# OpenReview forum: "Unveiling the Implicit Toxicity in Large Language Models"
_EMNLP/2023/Conference — EMNLP 2023 Main_

### Official Review · Reviewer_CkEn · 2023-08-04

**Soundness:** 4

**Excitement:**

4: Strong: This paper deepens the understanding of some phenomenon or lowers the barriers to an existing research direction.

**Paper Topic And Main Contributions:**

The work talks about implicit toxicity in language models by generating implicit toxic sentences using LLMs and later fine-tuning classifiers on the generated dataset.

**Questions For The Authors:**

I'm curious to know if the authors tried looking into why increasing the model size increases the toxicity of the model as shown in 4.6?

**Reasons To Accept:**

+ Using the generative capabilities of LLMs to generate toxic content.
+ Exhaustive experiments to demonstrate the effectiveness of the attacks.
+ Demonstrating that fine-tuning classifiers on the annotated dataset improves their performance.
+ Exhaustive ablation study showing the importance of various components.

**Reasons To Reject:**

- The authors frame their work as "attacks" but do not explicitly specify the constraints around these attacks. For example, in adversarial attacks works, there is a clear description of the settings where the attack would be successful and more importantly where it would fail, and the access needed for the attack -- back box, white box, etc., so it would be good to explicitly state the assumptions made, if any.

**Reproducibility:**

3: Could reproduce the results with some difficulty. The settings of parameters are underspecified or subjectively determined; the training/evaluation data are not widely available.

**Reviewer Confidence:**

3: Pretty sure, but there's a chance I missed something. Although I have a good feel for this area in general, I did not carefully check the paper's details, e.g., the math, experimental design, or novelty.

---

> ### Author Rebuttal · Authors · 2023-08-28
>
> > 1. The authors frame their work as "attacks" but do not explicitly specify the constraints around these attacks. For example, in adversarial attacks works, there is a clear description of the settings where the attack would be successful and more importantly where it would fail, and the access needed for the attack -- back box, white box, etc., so it would be good to explicitly state the assumptions made, if any.
>
> Our method belongs to black-box attacks without requiring the gradient information of the classifiers.
>
> Since we introduce an existing toxicity classifier $P$ during the online RL stage, our method requires a considerable number of queries to the toxicity classifier $P$ to make the attack successful.
>
> > 2. I'm curious to know if the authors tried looking into why increasing the model size increases the toxicity of the model as shown in 4.6?
>
> Larger models have a greater capacity to absorb diverse linguistic features and extralinguistic knowledge during pre-training, which is important for expressing toxicity implicitly (Figure 6). Consequently, they can achieve a higher attack success rate.
>
> To further demonstrate this, we manually inspect the generated responses to compare different models' usage of linguistic features for expressing toxicity implicitly. We randomly sample 50 queries and obtain corresponding generated toxic responses from a RL fine-tuned 1.3B / 13B model, respectively. We then label the linguistic features used in each response.
>
> We report the average feature number used in each response and the distribution of the linguistic features for conveying implicit toxicity.
> From the results shown in the following table, we can see that larger models can combine more diverse linguistic features, thereby leading to more implicit toxic responses.
>
>
>
> | Model Size | Avg. Feature Number | Sarcasm | Circumlocution | Euphemism | Antithesis | Metaphor | Rhetorical Question |
> | ---------- | ------------------- | ------- | -------------- | --------- | ---------- | -------- | ------------------- |
> | 1.3B       | 0.89                | 65.0%   | 22.5%          | 10.0%     | 0.0%       | 2.5%     | 0.0%                |
> | 13B        | 1.40                | 34.9%   | 27.0%          | 19.0%     | 12.7%      | 3.2%     | 3.2%                |

---

### Official Review · Reviewer_j5Tw · 2023-08-11

**Soundness:** 3

**Excitement:**

3: Ambivalent: It has merits (e.g., it reports state-of-the-art results, the idea is nice), but there are key weaknesses (e.g., it describes incremental work), and it can significantly benefit from another round of revision. However, I won't object to accepting it if my co-reviewers champion it.

**Paper Topic And Main Contributions:**

The paper explores LLMs' ability of generating implicit toxic outputs that are difficult to be detected by existing toxicity classifiers. To induce implicit toxicity in LLMs, the paper proposes a reinforcement learning based attack method, which can optimize LLMs with a reward that prefers implicit toxic outputs to explicit toxic and non-toxic outputs. Finally, the paper demonstrates that finetuning toxicity classifiers on the annotated examples generated from proposed attack method can enhance the abilities of detecting the implicit toxic outputs of LLMs.

**Reasons To Accept:**

The paper explores a safety risk of LLMs, namely the ability of generating implicit toxic outputs that are difficult to be detected by existing toxicity classifiers. A reinforcement learning based method is proposed to furthur induce implicit toxicity in LLMs by optimizing the reward that prefers implicit toxic outputs to explicit toxic outputs and non-toxic outputs. Extensitive experiments demonstrate the effectiveness of proposed method. The paper also shows that existing toxicity classifiers can be enhanced by utilizing proposed attack method.

**Reasons To Reject:**

Thougn proposed method shows a good performance on inducing implicit toxicity of LLMs, the method needs to collect two datasets. Therefore, it is better to show the effect of collected datset scale for demonstrating the efficiency of proposed method.

More ablation studies can be done e.g. the effect of hyperparameters \alpha. The author also can provide more comparison results without using reinforcement learning.

**Reproducibility:**

4: Could mostly reproduce the results, but there may be some variation because of sample variance or minor variations in their interpretation of the protocol or method.

**Reviewer Confidence:**

4: Quite sure. I tried to check the important points carefully. It's unlikely, though conceivable, that I missed something that should affect my ratings.

---

> ### Author Rebuttal · Authors · 2023-08-28
>
> > 1. Thougn proposed method shows a good performance on inducing implicit toxicity of LLMs, the method needs to collect two datasets. Therefore, it is better to show the effect of collected datset scale for demonstrating the efficiency of proposed method.
>
> Firstly, we study the size effect of the corpus for supervised learning. We report the reward value, the annotated toxic probability, and the average attack success rate over five widely-adopted toxicity classifiers. From the results, we can see that increasing the data size from 100 to 10,000 consistently improves the model's ability to express implicit toxicity.
>
> | Data Size | Reward | Annotated Toxic Prob. | Avg. Attack Success Rate |
> | --------- | ------ | --------------------- | ------------------------ |
> | 100       | -0.33  | 18.77                 | 61.38                    |
> | 1,000     | -0.217 | 47.74                 | 65.81                    |
> | 5,000     | 0.33   | 52.73                 | 69.63                    |
> | 10,000    | 0.35   | 54.02                 | 76.67                    |
>
>
> Secondly, we study the size effect of the corpus for reward model training. We report the accuracy of the reward model. The reward model accuracy consistently increases as the corpus size grows from 200 to 20,000, which is accordant with previous findings of the scaling law for reward model accuracy [1].
>
> | Data Size | Accuracy |
> | --------- | -------- |
> | 200       | 65.8     |
> | 2,000     | 75.9     |
> | 10,000    | 83.8     |
> | 20,000    | 86.0     |
>
> > 2. More ablation studies can be done e.g. the effect of hyperparameters \alpha.
>
> We conduct ablation studies to investigate the effect of the hyperparameter $\alpha$.
>
> As we can see, increasing $\alpha$ within a reasonable range improves attack success rates while keeping a comparable toxic probability. However, too large $\alpha$ results in a substantial decrease in toxic probability since the existing toxicity classifier $P$ mainly introduces the non-toxic signal while lacking the implicit toxic signal.
>
> | $\alpha$ | Reward | Annotated Toxic Prob | Avg. Attack Success Rate |
> | -------- | ------ | -------------------- | ------------------------ |
> | 1        | 2.23   | 59.49                | 83.56                    |
> | 5        | 2.47   | 58.84                | 86.58                    |
> | 20       | 0.89   | 25.81                | 86.42                    |
>
> > 3. The author also can provide more comparison results without using reinforcement learning.
>
> As illustrated in Section 4.2, we have conducted experiments with two non-RL methods, including supervised learning (SL) and supervised learning-rerank (SL-R).
> We further add a recently proposed method called Chain-of-Hindsight (COH) [2], which is also capable of learning from ranked data through supervised learning. Specifically, while RL learns a reward model on the ranked data and optimizes the policy model based on this reward, COH directly turns the ranked data into a sequence and adopts standard supervised learning. For example, given a context $x$, an implicit toxic response $y^w$, and an explicit toxic response $y^l$, COH constructs a training example as follows:
> ```plain
> Query: $x$
> An implicit toxic response: $y^w$
> An explicit toxic response: $y^l$.
> ```
> During inference, the model is instructed with "An implicit toxic response:" to generate desired outputs.
>
> The results in the following table indicate that both SL-R and COH show improvements over the SL baseline, while RL achieves the best performance.
>
> | Method | Reward | Annotated Toxic Prob. | Avg. Attack Success Rate |
> | ------ | ------ | --------------------- | ------------------------ |
> | SL     | 0.35   | 54.02                 | 76.67                    |
> | SL-R   | 1.01   | 55.23                 | 79.05                    |
> | COH    | 0.67   | 58.52                 | 77.95                    |
> | RL     | **2.47**   | **58.84**                 | **86.58**                    |
>
>
>
> [1] Training a Helpful and Harmless Assistant with Reinforcement Learning from Human Feedback. arXiv 2022.
>
> [2] Chain of Hindsight Aligns Language Models with Feedback. arXiv 2023.

---

### Official Review · Reviewer_BA3Z · 2023-08-11

**Soundness:** 3

**Excitement:**

3: Ambivalent: It has merits (e.g., it reports state-of-the-art results, the idea is nice), but there are key weaknesses (e.g., it describes incremental work), and it can significantly benefit from another round of revision. However, I won't object to accepting it if my co-reviewers champion it.

**Paper Topic And Main Contributions:**

The paper proposes an attack method based on reinforcement learning to induce implicit toxicity in LLMs. Experimental results on five toxicity classifiers demonstrate that the attack performace can be improved through RL fine-tuning.



**Questions For The Authors:**

1.Would the method can be extend to multi-modality context, such as CLIP?


**Reasons To Accept:**

1.The paper is well-written.

2.The experimental results seems comprehensive and promising.

3.The illustration and analysis is easy to understand.


**Reasons To Reject:**


1.It would be better to provide the training time. How time consuming for the LLMs?


**Reproducibility:**

4: Could mostly reproduce the results, but there may be some variation because of sample variance or minor variations in their interpretation of the protocol or method.

**Reviewer Confidence:**

2: Willing to defend my evaluation, but it is fairly likely that I missed some details, didn't understand some central points, or can't be sure about the novelty of the work.

---

> ### Author Rebuttal · Authors · 2023-08-28
>
> > 1. It would be better to provide the training time. How time consuming for the LLMs?
>
> Our experiments are carried out on 8 A100 GPUs (80GB).
>
> The supervised learning / reward model training / reinforcement learning stage costs 3 / 2 / 4 hours, respectively.
>
> > 2. Would the method can be extend to multi-modality context, such as CLIP?
>
> Our framework can be extended to multi-modality contexts. For example, while we use LLaMA as the backbone of the policy model and the reward model in our experiments, it can be enhanced to become vision-language models (e.g., MiniGPT-4 [1]) by introducing a visual encoder. Notably, while vision-language models attract increasing attention, the safety risks of vision-language models are still under-explored. We leave this for future work.
>
> [1] MiniGPT-4: Enhancing Vision-language Understanding with Advanced Large Language Models. arXiv 2023.

---

### Official Review · Reviewer_PPqG · 2023-08-12

**Soundness:** 4

**Excitement:**

3: Ambivalent: It has merits (e.g., it reports state-of-the-art results, the idea is nice), but there are key weaknesses (e.g., it describes incremental work), and it can significantly benefit from another round of revision. However, I won't object to accepting it if my co-reviewers champion it.

**Paper Topic And Main Contributions:**

This paper discusses the potential safety risks of large language models (LLMs) to automatically generate implicit toxic text. It proposes a reinforcement learning-based attacking method to do so. The method trains a reward model to assign high reward to implicit toxicity, while lower reward to explicit toxicity and non-toxic, thereby finally optimizing the LLM with the PPO algorithm. Further, the paper validates that such implicit toxic text are hard to be detected by existing toxicity classifiers, while also having diverse linguistic features.

**Questions For The Authors:**

1. What does the range of fleiss kappa mean (edit Aug 23. to add some context. In line 233, "ranging from 0.58 to 0.72", how is this range calculated)?
2. Do the authors plan to release the dataset for public inspection and future use?

**Reasons To Accept:**

1. The paper shows that implicit toxic text can be generated from LLMs through a reinforcement learning approach to optimize the LLM to prefer implicit toxic text to explicit and non-toxic texts.
2. Experimentally, the paper showed that the generated text can trick existing classifiers, and may have potential safety risks.
3. The paper is well written and easy to follow.
4. The experiments are thoroughly conducted, the prompts, annotation guidelines are provided.

**Reasons To Reject:**

1. Methodology-wise, the paper is using the (nowadays) standard rlhf with a small tweak on the reward function to target against existing classifiers. This is definitely not something that would reject the paper, but a reason why it didn't appear too novel to me.

**Reproducibility:**

4: Could mostly reproduce the results, but there may be some variation because of sample variance or minor variations in their interpretation of the protocol or method.

**Reviewer Confidence:**

4: Quite sure. I tried to check the important points carefully. It's unlikely, though conceivable, that I missed something that should affect my ratings.

---

> ### Author Rebuttal · Authors · 2023-08-28
>
> > 1. Methodology-wise, the paper is using the (nowadays) standard rlhf with a small tweak on the reward function to target against existing classifiers. This is definitely not something that would reject the paper, but a reason why it didn't appear too novel to me
>
> The main novelty of our paper lies in unveiling a novel safety risk of LLMs, namely their ability to generate implicit toxic outputs that are exceptionally difficult to detect with existing toxicity classifiers. We demonstrate this novel safety risk via zero-shot prompting (Section 2) and a RL-based attacking method (Section 3).
>
> Regarding the novelty in method design, we propose to design a novel reward signal which prefers implicit toxic responses to explicit toxic responses for attacking state-of-the-art toxicity classifiers. Methodologically, the most related work is TOXIGEN, which introduces an existing toxicity classifier to penalize explicit toxicity during the decoding stage. However, they do not intentionally design a signal to steer the model towards implicit toxicity.
>
> > 2. What does the range of fleiss kappa mean (edit Aug 23. to add some context. In line 233, "ranging from 0.58 to 0.72", how is this range calculated)?
>
> It denotes the Fleiss' Kappa based on the annotation results of different models.
>
> Specifically, for each model, we recruit annotators to label the generated texts as toxic or non-toxic. We then use Fleiss' Kappa to measure the inter-annotator agreement of the annotation results of different models respectively.
>
> We present the detailed Fleiss' Kappa value in the following table. The results demonstrate a moderate to substantial inter-annotator agreement.
>
> | Model         | Fleiss' Kappa                |
> | ------------- | ---------------------------- |
> | GPT-3.5-turbo | 0.58 (moderate agreement)    |
> | SL LLaMA-13B  | 0.65 (substantial agreement) |
> | RL LLaMA-13B  | 0.72 (substantial agreement) |
>
> > 3. Do the authors plan to release the dataset for public inspection and future use?
>
> Yes, we will release our data and models for public use.

---

### Meta-Review · Area_Chair_kyM3 · 2023-09-18

**Recommendation:** 4

**Metareview:**

Paper uses RLHF techniques fine-tune LLaMA-13B to generate implicit toxic content ("toxic while not using explicit toxic words"). "Our findings suggest that LLMs pose a significant threat in generating undetectable implicit toxic outputs." Next they train a RoBERTA classifier on a new human-labeled dataset (to be released) of such data and show it can detect such data with much better performance.

Overall shows an interesting way to get around existing toxicity classifiers and is an important practical topic to highlight in the LLM space.

---

### Decision · Program_Chairs · 2023-10-07

**Decision:**

Accept-Main

**Comment:**

Paper uses RLHF techniques fine-tune LLaMA-13B to generate implicit toxic content ("toxic while not using explicit toxic words"). "Our findings suggest that LLMs pose a significant threat in generating undetectable implicit toxic outputs." Next they train a RoBERTA classifier on a new human-labeled dataset (to be released) of such data and show it can detect such data with much better performance.

Overall shows an interesting way to get around existing toxicity classifiers and is an important practical topic to highlight in the LLM space.